# Multimodal Information Processing and Associative Learning in the Insect Brain

**DOI:** 10.3390/insects13040332

**Published:** 2022-03-28

**Authors:** Devasena Thiagarajan, Silke Sachse

**Affiliations:** Department of Evolutionary Neuroethology, Max Planck Institute for Chemical Ecology, Hans-Knöll-Str. 8, 07745 Jena, Germany; dthiagarajan@ice.mpg.de

**Keywords:** sensory systems, olfaction, vision, mechanosensation, gustation, neuronal circuitry, multimodal integration, associative learning, mushroom body, lateral horn, central complex

## Abstract

**Simple Summary:**

Insect behaviors are a great indicator of evolution and provide useful information about the complexity of organisms. The realistic sensory scene of an environment is complex and replete with multisensory inputs, making the study of sensory integration that leads to behavior highly relevant. We summarize the recent findings on multimodal sensory integration and the behaviors that originate from them in our review.

**Abstract:**

The study of sensory systems in insects has a long-spanning history of almost an entire century. Olfaction, vision, and gustation are thoroughly researched in several robust insect models and new discoveries are made every day on the more elusive thermo- and mechano-sensory systems. Few specialized senses such as hygro- and magneto-reception are also identified in some insects. In light of recent advancements in the scientific investigation of insect behavior, it is not only important to study sensory modalities individually, but also as a combination of multimodal inputs. This is of particular significance, as a combinatorial approach to study sensory behaviors mimics the real-time environment of an insect with a wide spectrum of information available to it. As a fascinating field that is recently gaining new insight, multimodal integration in insects serves as a fundamental basis to understand complex insect behaviors including, but not limited to navigation, foraging, learning, and memory. In this review, we have summarized various studies that investigated sensory integration across modalities, with emphasis on three insect models (honeybees, ants and flies), their behaviors, and the corresponding neuronal underpinnings.

## 1. Introduction

Insects perform precisely controlled tasks in extremely small time scales to navigate ecological niches and therefore serve as an excellent system to study complex behaviors and their origins. Reliable cues from an external environment are critical for decision-making and most insect behaviors occur as a consequence of simultaneous input-consolidation through multimodal sensory channels. For example, a predatory robber fly processes olfactory, visual, and directional cues simultaneously before executing a calculated aerial attack on a potential prey [1]. Similarly, a pollinating bumblebee in flight receives an overload of sensory information from a colorful flower emitting attractive volatiles [2]. In both these examples, diverse cues from the surroundings translate to information and elicit meaningful responses from the insects. Such multimodal integration has also been investigated in the context of courtship, mating, fleeing, and feeding behaviors of insects. Owing to the ease of approach and experimental design in laboratory conditions, unimodal sensory processing has been studied in different insect models and their isolated functioning has been extensively investigated [3,4,5,6,7]. However, multimodal integration, behaviors that utilize multimodal cues and their neuronal workings are still open fields for exploration. New discoveries made in this field emphasize the role of multimodal integration in improving the decision-making ability of insects, by altering the speed and accuracy of directed responses to stimuli. Therefore, it warrants the importance of investigating sensory signals in combination than individually. To that end, several studies have adapted virtual reality arenas to provide insects with the control of their own flight environments. This technique allows for the design of ecologically relevant experiments to study visual behaviors in navigation [8] and the mechanism of color learning in honeybees [9]. Aside from bridging gaps in our understanding of complex brain functions in tiny insects, research in this field also provides a wealthy foreground to inspire modern engineering feats such as bio-inspired development of micro-robotic control architectures [10]. In this review, we report the progress of research done in the past few decades in multimodal sensory integration, with an emphasis on honeybees, ants, and flies. This review also addresses the different behaviors that are popularly studied as a consequence of multimodal integration—namely innate flight behaviors using tethered flight arenas and associative learning. Finally, with a brief anatomical description of the major sensory systems (special emphasis on the vinegar fly *Drosophila melanogaster*), we discuss the major players of sensory integration in the central nervous system of insects, the three important brain structures—the mushroom bodies, the central complex, and the lateral horn.

## 2. Sensory Processing and Perception across Multiple Modalities

All complex behaviors arise as a robust response to both internal states as well as external multimodal stimuli. In popular vertebrate models, the superior colliculus, and the cortex have been identified as the sites for multimodal integration [11,12]. Specific neurons called the bi- and tri-modal neurons show suprathreshold responses from more than one sensory input and perform sensory integration when the inputs are presented together [13].

Several studies on diverse insect models have reported behaviors where different channels of sensory inputs interact to generate an innate response. For example, during host-seeking behavior, the female mosquito *Aedes aegypti* utilizes more than just one channel of sensory signaling (CO_2_, visual and thermal cues) to locate its next blood meal [14]. The female bush cricket *Requena verticalis* exploits both visual and acoustic cues while orienting toward a potential male for mating [15]. In the popular vinegar fly model *D. melanogaster,* three sensory channels—taste, olfaction, and mechanosensation—provide combinatorial input to initiate feeding behavior [16]. Antennal neurons of the carabid beetle *Pterostichus oblongopunctatus* respond to both air humidity and temperature [17]. Different moth species also display robust sensory integration to mediate and induce feeding [18]. For example, the nocturnal hawkmoth *Manduca sexta* finds both odor and visual cues equally attractive, but requires the synergistic input of both modalities to elicit proboscis extension and feeding behavior [19,20].

Aside from immediate navigation, the presentation of bi- or tri-modal stimuli can also confer contextual significance. In the sensory scene of pollination, flowers resemble a billboard of cues that the insects use to learn the nectar availability of the plant [21]. The more the number of the cues that are paired with the reward, the quicker and more efficient are the foraging behaviors. These associations are stored either as short- or long-term memories in the brain of the insect and retrieved later while repeating foraging trips. Such experiences in the environment—desirable or adverse—are learned and retained in the brains of insects, often in association with the otherwise neutral sensory cue (e.g., odor, color, shape). In such a process of associative learning, the sensory information acts as an indicator to the potential outcome of the experience. This association is used at a later point in time to remember the earlier encounter and generate a suitable behavior—to avoid or to approach the stimulus. The sensory cue is called the conditioned stimulus (CS) while the reward or the punishment that accompanies it is called the unconditioned stimulus (US). The representation of the CS in the neural substrate undergoes modification after being paired with an experience to signify a different meaning, therefore coding for a different behavior than before. Such learned behaviors have been studied intensively for several decades in insects of different orders. Insects utilize both olfactory and visual cues to learn, but the strength of the learning varies greatly between the two modalities. Olfactory learning has been the subject of research for several decades and the underlying circuits in the brain have been elucidated in great detail [22,23,24,25,26]. During the training period, an odor is paired with a reward (sucrose solution) in an appetitive conditioning paradigm or a punishment (electric shock, bitter tasting compounds, heat puffs) in an aversive conditioning paradigm. After repetitive reinforcement of this association, the choice of the insect toward this odor is tested. When positively associated, the insect approaches the odor and when punished, the odor is avoided. Similar paradigms have also been used to study visual learning, where a distinct visual cue replaces the odor as the conditioned stimulus [3,7,27,28]. Colors within the visible spectrum of insects and patterns (e.g., shapes, stripes) are used to reinforce rewards and punishments and the learned behaviors are tested using conventional T-maze choice assays or tethered flight arenas. In this review, we will summarize the studies that report bimodal learning behaviors in three insect models and the neuronal substrates underlying them in the brain.

## 3. Bimodal Processing and Learning in Popular Insect Models

### 3.1. Honeybees (Apidae)

The honeybee has been one of the most sustained research models for several decades, owing to its eusocial construct of living and the huge repertoire of complex behaviors that comes with it. With a total of only 950,000 neurons [29], the remarkable cognitive capabilities of these insects and their neural correlates have been the subject of intrigue in the field of learning and memory [30]. Individual hive members consolidate multi-channel information within and outside the hive to perform sophisticated tasks specific to their rank. Starting from the early works of Nobel laureate Karl von Frisch in 1965, different ethological approaches have uncovered the role of bimodal integration in pollinators, especially in a plant-pollinator context [31,32,33,34,35,36,37]. The transition between the 20th and the 21st century saw the development of classical conditioning experiments in restrained honeybees, where the pairing of a unimodal olfactory or a visual stimulus with a reward or a punishment leads to robust associative learning of the stimulus. Since then, several studies have addressed olfactory and visual learning separately, establishing that honeybees are excellent learners of both these sensory modalities. The most popular paradigm was the proboscis extension response (PER) which was followed later by the tethered flight arena. In controlled laboratory conditions, the usage of both visual and olfactory stimuli in a single PER paradigm opened up possibilities to present a restrained honeybee with a combinatorial CS+ (i.e., the conditioned stimulus paired with a positive or negative reinforcer). The study was one of the first to demonstrate a positive interaction between the two modalities, where a previous training with a visual stimulus enhanced olfactory learning [38]. These findings contradicted the overshadowing effect of bimodal training reported in earlier works on foraging bees. The earliest reports of synergistic effects of color on odor learning indicated stronger memory formation when compared to isolated unimodal training [39,40,41,42]. In later years, compound learning as a paradigm also addressed the effect of simultaneous stimuli presentation during training. In such a paradigm, positive patterning defines that the US is paired with the compound stimuli and negative patterning means that the US is paired with the individual components. Such experiments show that ultraviolet light can be learned better than other wavelengths and can specifically interfere with the reinforcement of a reward with an olfactory-visual combination [43]. Notably, the honeybee was the first insect model demonstrated to solve both, positive and negative patterning tasks, that involved more than one modality [44,45]. Therefore, further physiological investigation of this behavior can help narrow down the specific neuronal correlates directly underlying sensory integration. It is also noteworthy that trained honeybees can exhibit cross-modal associative behavior wherein they can recall just the specific color that was present alongside an odor scent during reward reinforcement [46,47]. This effect, which was previously described in humans, providing evidence for an information transfer between different sensory modalities during active flight, thus simplifying the process of foraging and increasing fitness in the insects [37,48]. In the past decade, aversive conditioning paradigms using an electric shock as a punishment were employed to examine both olfactory and visual conditioning in similar assays, enabling more direct behavioral comparisons between them [30,31,38,42,43,49]. Such methods have also been widely replicated to identify the brain centers that underlie the learning behaviors of both modalities [50]. Being a robust research model that offers a large variety of multimodal behaviors to choose from, honeybees provide a large ground to investigate cognitive tasks that involve complex sensory processing, both within and outside the laboratory environment.

### 3.2. Ants (Formicidae)

The ant is another hymenopteran model that has received a lot of attention from ethologists and neurobiologists alike for exhibiting sophisticated social behaviors that are largely integrative in nature. Members of the ant family Formicidae are distributed across 12,000 different species and show large diversity in anatomy, physiology, and behavior. They occur in different terrestrial habitats in huge numbers and therefore offer great sampling access to perform population studies.

The foraging members of an ant colony are regularly faced with the challenge of finding food, which could be miles away from their nest, and then finding their way back home. Several studies have identified multiple navigational techniques employed by ants of different species and other hymenopterans by extension, to perform this task [51,52,53,54]. Path integration is the most important tool in the box as it helps the ant to update its current position relative to the nest [55,56]. By counting the number of steps taken in a direction and using the celestial compass for orientation, foraging ants are able to form trajectories back to the nest. In fixed terrestrial habitats with specific panoramic views, foraging wood ants of the species *Formica rufa* learn and encode diverse visual cues of the scene and recall them during navigation on a match-the-view basis, therefore providing evidence for very strong long-term memories [57,58]. In contrast to such rich visual habitats, desert ants belonging to the genus *Cataglyphis* perform more challenging navigational tasks in a featureless habitat with very few food rewards scattered very far from their nests, while also constantly under the risk of predation by robber flies and jumping spiders [59,60]. In such a scenario, desert ants are capable of utilizing consistent landmarks when making the trip back to the nest [61]. However, it is of recent consensus that a multimodal input is necessary to navigate to and from a potential food source [62]. Initially, olfactory cues were thought to influence only inter-colony communication and nest mate identification. Although visual cues render navigation possible, only olfaction can provide a chemical tracking of a potential food source. With only visual cues, the ant can reach within a few meters of the source, but when there is no odor plume to direct it further, it keeps altering its course until it encounters one. This final indicator of the food reward marks the foraging trip as a success and makes the ant remember this route for future trips [63,64]. Such studies have also revealed that the navigational strategies do not directly lead to the food, but to the location where the odor plume was first detected [65]. It has also been shown that certain environment-specific blends in the desert habitat evoked significant electro-antennogram (EAG) activity in the *Cataglyphis* ants and can therefore be detected by the animals [66,67]. These could potentially be used alongside landmarks for fine navigation. Furthermore, the same study also showed that these ants could use olfactory cues to remember the nest entrance when the visual representation of the same is inconspicuous, implying the role of chemosensation in homing behavior. In order to investigate if and how these two modalities are integrated, a later study published by the same group combined visual and olfactory cues to represent a landmark in the path of a foraging ant [68]. The combined cues enabled the ants to focus their search immediately after the first training trial, therefore saving a lot of time and energy while returning to the nest. When trained once to a combination of both sensory modalities, the ants still showed a strong recognition of the individual olfactory and visual cues in the test phase. Interestingly, upon extended the training with the combined cue, the responses to the single cues were broader and more ambiguous, implying that a stronger reinforcement of the combined cue was used as a more reliable indicator of the landmark. This behavior substantiates the efficiency of processing multisensory cues in the learning of landmarks during difficult foraging tasks.

The remarkable integrative behaviors of foraging ants in the field have been frequently used to understand the significance of complex cue computation. However, the mechanistic framework supporting these behaviors are yet to be fully understood, owing to the poorly elucidated neural circuitry in the ant brain. Although a fair bit of similarity exists between the anatomy of the ant brain and the bee brain, functional and physiological experiments on live ants continue to be a technical challenge, especially when the aim is to establish ecologically relevant paradigms. However, the last ten years have also seen the development of genetic tools including the use of CRISPR-Cas9 method to generate transgenic ants [69,70,71]. These techniques are aimed to manipulate and genetically trace the olfactory centers while observing the consequent effects on the social behaviors, thereby opening up a new portal for in-depth exploration into the brain of the ant.

### 3.3. Flies (Drosophilidae)

The reputation of the vinegar fly *D. melanogaster* is beyond formidable. The ease of genetic manipulation, amenability to different experimental approaches and rapid, affordable upscaling makes it one of the most indispensable basic research model system worldwide. A century’s worth of work has been done to understand various physiological and behavioral pathways of the insect brain, enabling scientists to use it as a template to draw countless parallels to the more complex vertebrate’s systems. Therefore, it is of no surprise that when multisensory behaviors began garnering attention, neurobiologists turned to the humble fly for answers [72].

Decades of fly research have focused on discrete behaviors that arise as a consequence of sensory input, with special attention given to vision and chemosensation, while the basis of multimodal mechanisms is still a nascent field of study. The most obvious incidences of multimodal interactions are reported in fly-feeding behaviors. While food-derived odorants provide the maximum input for the tracking of a potential food-source by the fly, several other important features such as taste, texture, color, temperature, and wind direction are also received and processed during the decision-making process. A synchronous addition of a mechanosensory and an olfactory cue to the taste stimulus enhances the proboscis extension response (PER) in flies and initiates feeding [16]. Wind directions indicating the odor plume trajectory and visual input via the optic flow are vital for navigation in the wild [73]. Contributions from different sensory modalities are essential for behaviors such as egg laying, where picking a suitable substrate would ensure the safety of the eggs and sustenance for the developing larvae. Alongside parasitoid- specific odor cues, *D. melanogaster* females also utilize visual cues to detect the presence of the wasp, which activates a signaling pathway to suppress egg laying [74]. Sensory integration also plays a major role in the communication of mating signals during the courtship ritual of animals [75]. A classic example is found in the male courtship behavior of the fly, where a combination of olfaction, gustation and vision is required for the male fly, not just to initiate the courtship with a virgin female, but also to sustain its sex drive and carry the courtship to completion and succeeding in copulation [76]. It has also been shown that the presence of a food odor can increase the salience of the male-released pheromone, cis-vaccenyl acetate (cVA), thereby preventing males from making futile attempts to court a mated female on feeding sites [77,78].

Combinatorial processing of sensory information is also seen in larval locomotion studies, where synergistic activation of mechanosensory and nociceptive neurons increased the likelihood of rolling, an escape behavior exhibited by the larvae [79]. Larvae can also compute integrative behaviors just before making the decision to turn [80]. Based on the CO_2_ levels, light intensity and the presence of attractive odors, *D. melanogaster* larvae use head-sweeps to scan the spatial gradients in the environment and linearly combine both olfactory and visual signals before executing a turn [81,82,83].

With the advent of the tethered flight era, many research groups have targeted the role of sensory integration during flight maneuvers [84]. Any behavior exhibited by a flying insect requires motor control with high spatio-temporal precision while also processing multimodal navigational cues that change in real time. Wing-beat analyzers coupled to a flight simulator provide a conducive set-up to study motor response behaviors to varying stimuli. Although different sensory organs relay different information about the environment, the flight control system integrates input from the halteres (i.e., modified hind-wings essential for flight) and the optic lobes, in a manner such that the motor response to the combined input is always greater than that elicited by just one modality [85]. Such summation models have also been observed in the integration of visual and mechanosensory stimuli during a turn-behavior executed by a tethered fly [86]. Closed-loop tethered flight experiments also show that visual feedback can increase olfactory acuity by regulating odor-motor responses of the fly [87]. Conversely, the same system has also been used to show modulation of visual salience by odor activity in a context dependent manner [88]. An attractive odor or the optogenetic representation of one can be used to reverse the aversive nature of a small object in the fly’s visual field [89,90]. In the wild, such inter-modality regulation systems can narrow down search behaviors and greatly increase foraging efficiencies. The ease of stimulus presentation in the tethered flight system coupled with recent advancements in optogenetic neuronal control provides a strong foundation to explore the circuit dynamics of multi-sensorimotor responses in the fly model.

The first instance of using more than one kind of stimulus in a learning paradigm also happened in the tethered flight simulator. When two different types of visual stimuli—colors and patterns are used as a CS to be associated with a heat stimulus—flies show robust compound learning, with equally strong associations being produced for both stimuli individually [91]. The study specifies that flies can acquire, store, and retrieve the two stimuli separately and also as a compound. A similar paradigm was then used to study cross-modal integration in flies, where instead of two stimuli of the same modality, the CS involved a combination of an odor and a visual pattern [92]. The study performs two important experiments that postulated possible information transfer between the two sensory modalities: In the first experiment, the flies were trained to unimodally associate a heat reinforcement to an odor of low concentration and a visual pattern stimulus (both elicited very low learning responses on their own). In such a scenario, a bimodal conditioning that consisted of the two stimuli paired with the heat reinforcement produced stronger learning performances, both when retrieved as a compound memory and as individual components. This observation further proves the principle of inverse effectiveness, where a weak memory reinforcement can be amplified using a cross-modal percept [93]. In the second experiment, a combination of both stimuli was provided simultaneously for “sensory preconditioning” after which each sensory modality was paired with the heat reinforcement individually. In the testing phase after such a preconditioned training paradigm, even retrieval with the non-reinforced stimulus produced a robust learning response, signifying a very strong cross-modal transfer of memory. Aligning with the experiments done in honeybees, these observations were pivotal in the understanding of bimodal information transfer that occurs during operant conditioning, especially in a situation of sensory deficit, where a different source of input reinforcing the same consequence can greatly aid in quick decision-making.

Every sensory element that constitutes a context holds weight in how the experience is remembered and can be used to retrieve the memory at a later point in time. A recent study [94] utilized specific components of the “Tully T-maze”, a two-choice learning paradigm [95], to illustrate this concept in *D. melanogaster*. This included the replication of specific aspects (except the US) of the aversive training paradigm onto the testing phase, such as the color of the light, the temperature of the chamber and the input of visual and mechanosensory cues from the copper coil that conducts the electric shock. The study aimed to understand the substrates of aversive conditioning and long-term memory (LTM) generated thereafter. The findings show that flies can perform context-based multimodal integration in response to an aversive learning experience. They use this information as a basis for forming long-lasting memories retrievable even after 14 days. During the testing phase, when replicating the context in which the reinforcement (here: an electric shock) was delivered, a significant long-term memory was formed immediately after the first training trial. Contradictory to older studies, targeted blocking of the protein consolidation did not impair this behavior in flies, clearly indicating that such a context-dependent LTM (cLTM) does not require protein synthesis. However, when the copper grid or the visual context of reinforcement was removed from the testing phase or when the perception of visual input was genetically inhibited, the cLTM was significantly abolished implying the importance of vision in the retrieval process. The importance of an encoding context in enabling efficient memory retrieval is often described in psychology and observed in complex vertebrate models, including humans. However, extensive physiological and molecular work are required to pinpoint the neural substrates that can relay information transfer between distinct sensory modalities.

## 4. Neuronal Substrates and Brain Centers Underlying Unimodal and Multimodal Processing

Although several behaviors are studied as a function of unimodal sensory processing and years of research have identified the importance of singular modalities in the execution of these behaviors, specific centers exist in the insect nervous system that form a close network between different sensory neuropils and sample multimodal inputs continuously. The information flow between these brain centers largely depends on the nature of the behaviors. The architecture of the insect nervous system is designed to perform both instantaneous maneuvers that bypass the central brain, controlled by the body ganglia as well as complex learned behaviors that require higher brain structures. Such a design is called a decentralized organization [96]. For example, most reflexive motor functions rely solely on the ventral nerve cord (VNC) and behaviors such as escape or flight involve its direct control, and may not involve the central brain [97,98]. Other slower sensory loops involving odor processing or visual guiding are mediated via their respective neuropils in the brain from the periphery. After the higher centers receive additional information from other sensory pathways such as context, experience, reward or punishment, signals are sent down through the descending neurons to the VNC to coordinate motor neuron responses and to elicit a corresponding behavioral output. While variation is seen in the number of neurons and the size of specific centers among most insects, the basic template of the organization is astonishingly conserved. Therefore, multimodal integration sites can be largely generalized and a lot can be learned from investigating the brains of different insect models.

The insect brain is made of two major ganglion groups—the supraoesophageal and the suboesophageal ganglia. In different insects, the supraoesophageal ganglia can be of different sizes and consist of different neuron densities, but show a conserved division into three regions—the proto-, deuto-, and the tritocerebrum. The protocerebrum consists of the optic lobes and higher processing centers such as the mushroom bodies (MB), the lateral horn (LH) and the central complex (CX). The antennal lobes and the antennal mechanosensory motor center make up the deutocerebrum. The sense of taste and other functions of the insect mouth parts are linked to the tritocerebrum and the suboesophageal ganglion [96,99,100]. Aside from processing only unimodal input, some primary sensory centers have also already been shown to respond to stimuli of other modalities as well. For example, the extensive connectome of the *D. melanogaster* larva revealed that the local interneurons of the antennal lobe receive input from unidentified non-olfactory sensory neurons, possibly gustatory in origin [101]. Recent work has also shown the converse, where gustatory receptor neurons respond to odors to regular proboscis extension reflex which serves as a readout for feeding behavior [102].

Advances in the genetic toolkit—especially the tissue specific binary expression system have recently made possible, the mapping of the intricate neuronal layers in the brain. For instance, the Gal4-UAS system has been widely utilized to mark and manipulate specific cells in an organism. The yeast-specific Gal4 protein is placed under the control of a native promoter while the UAS controls the expression of the introduced gene. When expressed together, the Gal4 targets the expression of the gene downstream of the UAS to specific cells marked by the native promoter, thereby ensuring cell-specific expression of desirable markers or genetic mutations [103]. When paired with optogenetic tools, the expression of light-sensitive ion channels downstream of the Gal4 allows for light-induced activation of individual neurons or a population of neurons [104]. The more recently developed split-Gal4 technique is a valuable addition to this repertoire, where the components of the binary expression system are initially split in two, but reconstitute when expressed in overlapping cell groups and drive the expression of the downstream fluorescent reporter [105]. Hundreds of *D. melanogaster* specific split-Gal4 lines have been developed to mark sparse neuronal populations and are used in conjunction with synaptic targeting systems to generate a comprehensive connectome of the adult brain [106,107,108,109,110]. With special emphasis on the *D. melanogaster* model, this section addresses the primary sensory systems and the major multi-sensory integration centers—namely the MB, the CX, and the LH.

### 4.1. The Insect Olfactory System—A High-End Chemosensor

Chemical cues present in the environment provide crucial information to the survival and thriving of all organisms. From food foraging to identifying potential mates, insects depend on the detection of chemosensory signals for all their behaviors. Chemosensory organs of insects spanning across different ecological niches have evolved with varying orders of sensitivity and remarkable structural modifications to adapt and serve different functions [111,112]. In a vast majority of insects, the antennae and the maxillary palps are the major organs associated with olfaction, while the sense of gustation heavily relies on the proboscis and the labial palps [113]. The olfactory system of the vinegar fly *D. melanogaster* has been studied extensively to understand the path of chemoreception of odor molecules, i.e., from the antennal apparatus on the periphery to the higher centers deep within the central brain where complex behaviors originate (Figure 1A). Most of the following anatomical description of the olfactory system can be considered as a representative of other insects, except when specified otherwise. Fine, hair-like structures called sensilla housing olfactory sensory neurons (OSNs) are distributed on the antenna, acting as the primary sites of odor reception [6]. Odor molecules enter the sensilla through pores on the walls and after interacting with odorant binding proteins (OBPs) in the aqueous medium, they reach the dendrites of the OSNs. Each OSN possesses specialized olfactory receptors (ORs) and/or ionotropic receptors (IRs) that bind to specific odor ligands. This binding initiates a depolarization of the corresponding OSNs, leading to the generation of action potentials [114,115,116]. Single sensillum recordings monitor these generated action potentials and identify receptor-ligand interactions of hundreds of chemical compounds [117]. Once detected by the OSNs, odor information is then taken to the primary brain center of the olfactory pathway, the antennal lobes. Densely packed neuropil units called glomeruli in the antennal lobes act as the first synaptic site for the OSNs. These discrete glomeruli vary in number in different species, but follow a fundamental organization. Most glomeruli receive input from only a single type of OSNs and the evoked activity patterns in the antennal lobe confer valence and identity to the odors [5,118,119]. Local interneurons (LNs) and projection neurons (PNs) further innervate the glomeruli and are the next major players in the odor computation process. They either convey information between the glomeruli or wire them to higher centers such as the MB and the LH, which are parallel higher processing centers. The MBs have been investigated extensively and are seen as the center for learning- and memory-associated behaviors [25,26,120]. They receive input from many uniglomerular projection neurons (uPNs) and a few multiglomerular projection neurons (mPNs) from the antennal lobes, which is then redistributed across a large number of intrinsic neurons called the Kenyon cells (KCs) [106]. The parallel axons of the KCs form the different lobes of the MB (γ, α′/β′, and α/β), each lobe retaining odor associative functions of its own. Mushroom body output neurons (MBONs) connect with KC synapses to largely mediate behavioral output and communicate via axons to neuropils outside the MBs such as the LH. This topic is further explained in more detail in 4.4. The LH, by itself, is known to produce responses to both innate and learned odor stimuli as they receive input from both the uPNs and the mPNs of the antennal lobe and the MBONs [121]. Owing to the large spectrum of odor-guided behaviors that originate from their functionality, the LH is a major focal point of interest in recent studies.

### 4.2. The Insect Visual System—A Thousand Tiny Eyes Working as One

Visual cues are perceived in many different forms—shapes, patterns, colors, and contrasts—by the compound eye and are utilized by insects for guiding various behaviors. Differently adapted insects possess varying visual acuity that is more suited to their ecological niches and behavioral preferences (predatory, social/solitary, nocturnal/diurnal/crepuscular). The compound eyes are called so because of the arrangement of numerous small eye units termed as ommatidia “compounded” together. The number of ommatidia can vary anywhere between 10 and 30,000 and their structural organization can differ, catering specifically to the needs of the insect [122]. Although we have not yet fully understood how visual perception transpires objectively, a fair bit of research has been done to decode the physiological underpinnings of this sensory modality by studying color perception, motion recognition, and polarized light detection as a function of the eye apparatus of insects. In *D. melanogaster*, the retina on the periphery contains over 800 ommatidia, each of which has eight photoreceptors (R1-R8). The roles of these eight photoreceptors have been studied extensively using electrophysiology and functional imaging experiments. While R1-R6 aid in the formation of images and vision during reduced light conditions, R7 and R8 are involved in color perception and detection of polarized light [123,124,125]. Although not directly required for color vision, recent studies have shown the contribution of photoreceptors R1-R6 in it [126]. Spectral information is derived at the very first synapse, closely resembling the vertebrate retina [127]. Specific rhodopsins provide spectral sensitivity to different wavelengths, with *rh3* showing a peak to short ultraviolet (short UV), *rh4* to long UV, *rh5* to blue, and *rh6* to green [128].

The *D. melanogaster* visual apparatus is home to over 60% of the total neurons in the entire brain. Therefore, it represents the largest sensory neuropil in the fly nervous system [129]. It shares striking similarities in organization and information flow with the olfactory apparatus [130]. The topographical arrangement consists of a retina on the periphery and four optic neuropils—the lamina, medulla, lobula, and lobula plate (Figure 1B). Axons arising from R1-R6 project to the lamina while those of R7 and R8 project to the medulla [131]. The medulla is the most complex optic neuropil and consists of approximately 100 different cell types, distributed between ten layers. The distal medullary region receives external input and wires it to the proximal region, which furthers the computation of visual information. Transmedullary neurons (Tm, TmY) connect the medulla to the lobula [125], which has 800 columns organized into six layers. Columnar neurons and the tangential (i.e., tree like) neurons of the lobula receive input from large visual fields similar to the medulla. Visual motion, direction-selective light inputs, and figure-ground discriminations are processed by the projection neurons of the lobula plate [132]. Their role in motion detection is extensively reviewed in this article [133]. This consolidated information from the lobula and the lobula plate is then transmitted to the ventrolateral protocerebrum (vlPr) that is located directly beneath the optic lobes. Most of the visual information from the optic lobes reaches the ellipsoid body (EB) of the CX via the anterior optic tubercle (AOTU) and the bulb (BU) neurons. Representation of the medulla and the lobula is spatially separated in the AOTU to filter out specific information from both these regions before conveying them to the CX. The role of the AOTU as a relay site between the visual periphery and the central brain is known to be conserved across many insect taxa, especially in the detection of polarized light and celestial navigation [134]. Thus, the preliminary visual information conveyed from the retina through each of the optic neuropils finally reaches distinct sites in the central brain, especially the CX, where contexts are derived to generate specific motor behaviors.

### 4.3. Gustation and Mechanosensation

As briefly mentioned before, through the bristles on the proboscis apparatus, the labial palps on the head and the receptors on the tarsi, the fly receives gustatory cues from tastants and other non-volatile chemicals [135]. The suboesophageal ganglion serves as the primary site in the brain where gustatory input is received [136]. The fly head, along with specific structures distributed across the length of the body, also contains mechanosensory and thermosensory receptors that convey information such as heat, pain, pressure, and vibrations (Figure 1C). Thermo- and hygrosensors are identified on the aristae of the antenna while tactile hairs on the legs and the thorax also serve as efficient mechanosensors. A specialized structure inside the antenna called the Johnston’s organ conveys distinct acoustic information, proprioceptive feedback, and other mechanosensory cues to the brain via the antennal nerve. The Johnston’s organ neurons (JONs) arborize throughout the brain region called the antennal mechanosensory motor center (AMMC), where specific cell types are known to respond to very selective mechanical stimuli [137,138]. Further downstream partners of the AMMC neurons include the wedge (WED) neurons, which receive diverse inputs from the AMMC that help in sensing the wind direction. This output then reaches the fan-shaped body (FB) of the CX via the lateral accessory lobe (LAL) [139]. However, the functional role of these neurons in regulating specific mechanosensory behaviors is only just beginning to be explored in detail.

### 4.4. The Mushroom Bodies

The French biologist Felix Dujardin made the discovery of the MBs in the hexapod brain in 1850. He named them so owing to the distinct shape of the bilaterally symmetrical calyx which is connected to the rest of the brain by a peduncle [140]. Basic insect research from the last two centuries has immensely advanced our knowledge we have on these specialized neuropil structures, especially the work done on specific models such as the honeybee *Apis mellifera*, the cockroach *Periplanata americana,* and the vinegar fly *D. melanogaster* [141].

The relationship between the antennal lobes and the MB is one that has been widely investigated in the processing of olfactory input. As mentioned previously, thousands of KCs are packed densely in the calyces of the MBs. The KC dendrites synapse with the incoming second-order PN axons from the antennal lobes, while the KC axons form the peduncle of the MB. Three different kinds of KCs extend in a parallel fashion to form the distinct MB lobes (α/β, α’/β’ and γ) where they proceed to form synaptic connections with a small number of (MBONs). The MBONs take inputs from the three lobes and project to other neuropils outside the MB such as the LH. Modulatory dopaminergic (DAN) neurons and octopaminergic neurons also innervate the MB lobes at specific subdomains and provide the substrates for aversive and associative learning behaviors. The detailed anatomy of the three MB lobes and their role in learning and memory can be found in these publications [23,26,96,106,110,142].

The main calyx receives most of its input only from the antennal lobes while the ventral accessory calyx receives input from the medulla of the optic lobe, the region that is known to process color and contrast information [143,144]. Using dextran dye injections at the primary sensory sites and genetic labelling techniques, multimodal sensory pathways from olfactory, visual and gustatory centers that project to the dorsal accessory calyx in *D. melanogaster* were identified [145], as was also previously observed in hymenopteran models [146,147]. The same dorsal accessory calyx was also recently shown to mostly integrate information received through two kinds of visual PNs, one set from the lobula of the optic lobe and the other from the posterior lateral protocerebrum [148]. The segregation of input occurring at the different calyces suggests a largely parallel functioning of the MB at the level of the KCs, similar to the concentric pattern of input segregation reported in honeybees and cockroaches [149,150].

The information from the KCs strongly converges onto very few (MBONs) which represent the next level of multisensory processing. Even in less commonly investigated models like the cricket, a single MBON was reported to respond to auditory, visual, and wind stimuli [151]. Studying the response properties of the honeybee MBONs revealed that about 42% of the total hits indicated visual sensitivity, while 32% indicated bimodal sensitivity to both light and odors. Only 9% of the MBONs showed unimodal odor sensitivity [152]. A more recent computation method done to map the connectivity between PNs and MBONs showed a similar distribution, with some MBONs predominantly assigned to non-olfactory modalities [106]. These findings are further supported by the role of MBONs in driving learned visual, olfactory, and bimodal behaviors, especially since they show distinct connectivity patterns with the DANs. An earlier study that employed the tethered flight set-up used MB mutants to elucidate the importance of this structure in the context generalization of visual learning [153]. However, studies that are more recent have directly implicated the role of the MBs in acquisition and retrieval of visual memories. Genetically blocking the output from the KCs to the MBONs abolished the ability of flies to acquire and retrieve aversive visual memories, showing that the MBs are also involved in the visual learning process. Another study also revealed a more interesting observation, where olfactory and visual inputs were coded by different sets of gamma KCs [143] while their respective associative memories (appetitive and aversive) are coded by overlapping, yet slightly distinct sets of KCs [154,155]. Additionally, KC dendrites of the main and the accessory calyx show sparse activation to gustatory cues and deficits in gustatory learning are seen when the function of the gamma lobes is impaired in mutant flies [156]. These findings conclusively illustrate the crucial role of the MBs in olfactory, visual, and gustatory associative learning and therefore not specific to a single sensory modality (Figure 1D).

Putative relationships have been identified between the MBs and the CX as early as the 1990s, with one study illustrating the common effect light had in the sizes of the calyces of the MB and the CX [157]. While several MBONs send their projections to the CX, the connections from the CX to the MB are more limited. The information from the MB to the CX is primarily utilized by the navigation control system to derive experience-based instructions upon which motor behaviors are executed and controlled. However, an interesting study using the tethered flight arena identified the role of the MBs in a memory-independent olfactory modulation of visual response that is essential for flight control. The same study also notes that the reverse scenario (visual modulation of an olfactory response) is not regulated by the MBs [158]. To our knowledge, this was the first instance where a cross-modal function of the MB was identified that was independent of learning or memory related circuits.

With the advent of high-resolution microscopy and EM-derived mapping of the MB neural substrates, detailed connectomics of the insect brain has become known [23,106,109,159]. The anatomy and the physiology of the MB neuropils along with its conserved architecture (now elucidated), provide great insight into preliminary multimodal integration that was previously not substantiated. Moreover, the generation of specific and sparse transgenic neuronal lines in the MB can now be used as a tool to manipulate and knockout different candidate neurons, thereby elucidating the specific roles of the MB neurons in regulating multimodal integration and their corresponding behaviors. Such explorative and descriptive studies can lead to the design of further behavioral paradigms, to fully exploit the system and reveal the workings of more complex behaviors.

### 4.5. The Central Complex

Along the midline of the insect brain lies a modular structure called the central complex (CX). While its discovery dates back as early as the mid-1800s, definitive descriptions of its organization across the entire insect order only came later. The actual term was coined in 1943 by Maxwelle Power while describing the nervous system of the vinegar fly [160]. In the same article, Power wrote that the CX is an important association center as it receives input from different parts of the brain. A detailed review on the evolution of the CX can be found here [161]. The CX consists of four neuropils—the protocerebral bridge (PB), the fan shaped body (FB), the ellipsoid body (EB), and the noduli (NO). Each of these structures show stereotypical inter-connectivity and are also well-connected with structures external to the CX that control motor output.

As a structure that processes information from both the environment as well as internal states such as hunger and mating status, the functions of the CX are diverse, ranging from guided locomotion to long-term learning and memory behaviors [162,163,164,165]. Compartments of the EB are involved in resolving spatio-temporal cues required for landing maneuvers, flight control, and orientation. The FB plays an important role in executing walking behaviors and negotiating barriers in a path. Neurons of the ventral FB are also known to encode airflow direction, required for orienting toward a stimulus [166]. In a study that investigates nociceptive perception in the fly model, it was shown that harmful stimuli including electric shock are coded as innate and conditioned responses in the FB [167]. Given the extensive usage of electric shock punishments in aversive learning experiments, this finding suggests putative communication between the MB and the FB. Both the EB and the FB have tremendous contribution from the visual centers, especially in using polarization cues and the sky compass, which are essential for navigation [168]. Most motor behaviors require feedback processing from the signals of the environment and the detection of the insect’s internal axis. A specific type of recurrent networks called the ring attractors were long thought to regulate directionality and heading behaviors, by sustaining a bump-like activity pattern, which is modulated after every turn or shift in direction. This model was also physiologically proven [169,170].

The reception and processing of varying sensory inputs to generate an appropriate motor output is the most prominent function of the CX and has been characterized in several insect models such as moths, honeybees, ants, flies, and cockroaches [161,163,171,172]. Predictably, studies that utilized CX mutants show a wide range of motor deficits, including inability to localize targets and initiate walking. A recent study proposed a model of the steering circuit in the CX that utilizes a large array of olfactory, visual, and mechanosensory cues to relay context-dependent motor guidance, further strengthening the function of CX as the navigational switch of the insect nervous system [31].

Although the MBs have been long implied as the center for learning behaviors, the role of the CX in spatial learning has also been of interest. Goal-directed responses controlled by distinct layers of the FB are shown to aid the MBs in maintaining classical memories while also allowing for flexibility, as the fly’s situation changes [173]. Specific neurons in the EB are required for visual place learning (using distinct visual cues to direct navigation) and silencing the input to the EB hugely impairs this behavior [174]. A connection between the MBs and the CX was identified in honeybees with regard to spatial learning behaviors, where the CX was involved in goal-directed responses and the MBs performed associative behaviors. Targeted manipulation of specific neurons in either structure resulted in clear impairment of the behavior [50]. Such connections have also been reported in *D. melanogaster* with a single MBON receiving input directly from the CX and several MBONs exhibiting direct connections to the fan-shaped body [106]. Mutant animals that had a defective CX also showed reduced olfactory learning performance [175]. Impaired ipsi- and contralateral gustatory habituation was observed in *no-bridge^KS49^ (nob)* mutants that have a disturbed protocerebral bridge, implying that some of the communication between the two brain hemispheres is mediated by the CX [176]. These observations support an information transfer between the MBs and the CX, so that learned information, internal state, and previous sensory experiences can be used to generate a sophisticated and rapid motor function [177,178].

With a myriad of functions that encompass olfactory, visual, gustatory, and mechanosensory modalities, the mechanistic framework of the CX remained elusive for several years, with new discoveries on its neuronal architecture and connectivity being continuously brought to light (Figure 1D). Moreover, the ability of the CX structures to process highly variable (egocentric and geocentric) information from the environment and to generate a coherent motor output for navigational purposes signifies its complex computational potential, making it an excellent structure to study it as the neural substrate of sensory integration. Exploiting the genetic tools available in *D. melanogaster* to label and manipulate single neurons can allow future work to elucidate the role of each CX substructure and their neuronal connections to other brain centers, in the generation of behaviors influenced by multisensory inputs.

### 4.6. The Lateral Horn

The LH is another part of the protocerebrum in the insect brain, which has garnered major research interest in the last two decades, especially in the *D. melanogaster* model. It is one of the two major higher olfactory processing centers and is often linked to the control of innate olfactory behaviors by providing a biological context to the odor cues, as it receives stereotypical axonal projections from the antennal lobes [179]. Detailed reviews on the role of the LH in olfactory processing can be found here [121,180,181]. Different regions of the LH are also known to respond to odors that have positive and negative hedonic valences [182]. It was also shown that LH neurons (LHNs) are more broadly tuned when compared to their input neurons and show different responses to odors belonging to different chemical classes [183]. A potential role for the LH in learned olfactory behaviors was first proposed in context-dependent long-term memory formation, indicating more diverse functions for the structure than previously described [94].

The first study to generate an array of neurogenetic reagents that labelled LHNs utilized the split-Gal4 technique that splits the components of the binary expression system, which reconstitute when expressed in the same cell and drive the expression of the downstream fluorescent reporter in only a sparse subset of neurons [108]. An astounding 2444 lines were generated and screened using this technique, which were further filtered for maximum efficiency to over a 100 lines. The lines in this repertoire further lead to the identification of LH input neurons (LHINs), LH output neurons (LHONs), and LH local neurons (LHLNs).

Studying the LHINs revealed that the LH receives input also from regions other than the antennal lobes, such as the auditory and mechanosensory system, the gustatory system and the lobula of the visual system [108]. This study emphasized that the ventral region of the LH is the site of multimodal convergence, with major inputs coming in from the auditory and the mechanosensory systems. Some neurons showed a strong presynaptic signal in the VNC, suggesting the transfer of putative pheromone-based or mechanosensory inputs from the VNC to the LH [184,185,186,187]. Another multimodal connection was also proposed in a specific LHON (AV2b1/b2), which potentially receives both visual and temperature input [108]. The position of this neuron is close to a site of several olfactory inputs, suggesting that it could be conferring visual context to an odorant. This observation is further supported by the tethered flight experiment done in the same study, where the optogenetic activation of the LHNs drove the flies to move towards a visual stimulus [108]. In a different study, it was shown that activation of higher-order LHONs (so-called VLPn) induced contralateral inhibition in the LH, influencing navigational responses of a fly to a gradient of odors across the two antennae [188]. Interestingly, these LH neurons have also been shown to receive inputs from other modalities, therefore providing further evidence for the LH being a site of multimodal convergence. It can also be hypothesized that the LH could be an intermediate site between multimodal input and motor output [189,190,191]. Specific connections were revealed between the LHONs and the MBONs implying a potential cross-modulation of associative behaviors between the MB and the LH and can be used to understand their combined role in eliciting complex behaviors [23,155]. A recent study done on the turning behavior of *D. melanogaster* larvae confirms functional connectivity between the LH and MB pathways and illustrates a mechanism by which innate and learned valences interact. A distinct sub-type of neurons called convergence neurons (CNs) is described in this study, which is not only activated by the attractive LH pathway, but also receives excitatory and inhibitory input from MBONs that encode aversive valence [192].

In summary, in spite of receiving the majority of input from the olfactory PNs, the LH also collects inputs from centers processing other sensory modalities, such as vision, sound, temperature, mechanosensation, and gustation (Figure 1D). It also receives input from the learning center via the MBONs. This postulates two potential roles for the LH: the multimodal repertoire of information reaching the LH can provide a context upon which the biological valence for an odor is built, and that the LH serves as a center for processing non-olfactory input that promotes downstream control of motor behaviors. Both roles emphasize the capacity of the LH in eliciting complex and context-dependent behaviors in insects and therefore provide a rich foundation for future work that can identify the underlying multimodal neuronal networks.

## 5. Conclusions

Single sensory modalities have been studied in great detail by elucidating the reception (including identification of the sensory receptors), the processing mechanism, the key neuronal players involved as well as the behavioral output. However, in the wild (not in the lab!) an animal is hardly exposed to a sparse environment, but rather faced with multimodal sensory impressions which need to be detected and consolidated in a context-dependent manner. Simultaneous monitoring of multiple modalities in the insect brain used to be an uncharted territory, but recently gained increasing attention due to the development of high-end genetic tools and imaging techniques. In light of recent studies, previously considered as a site that only regulated innate behaviors, the LH has been shown to play different, more diverse roles in processing multisensory input and learned associations. This observation renders a more plastic nature to the LH and makes it an interesting candidate to study sensory integration.

Virtual reality inspired artificial arenas, which are built in order to achieve similarity to ecological niches, can also provide crucial insight into real-time understanding of complex insect behaviors. Such integrative approaches are essential to fully comprehend the mechanistic origins of behaviors. These technological advancements provide an edge for the insect models over complex vertebrate systems. Some key questions in the field include how different sensory systems modulate each other, what are the underlying neuronal mechanisms, and how multi-layered information processing can be deciphered using simple insect-model templates. Feedback of multisensory integration on primary sensory centers is also not well established, which is vital in the understanding of temporally regulated motor behaviors. This information will serve as a building block in the booming field of insect behavioral studies, which can also be translated across diverse taxa.

## Figures and Tables

**Figure 1 insects-13-00332-f001:**
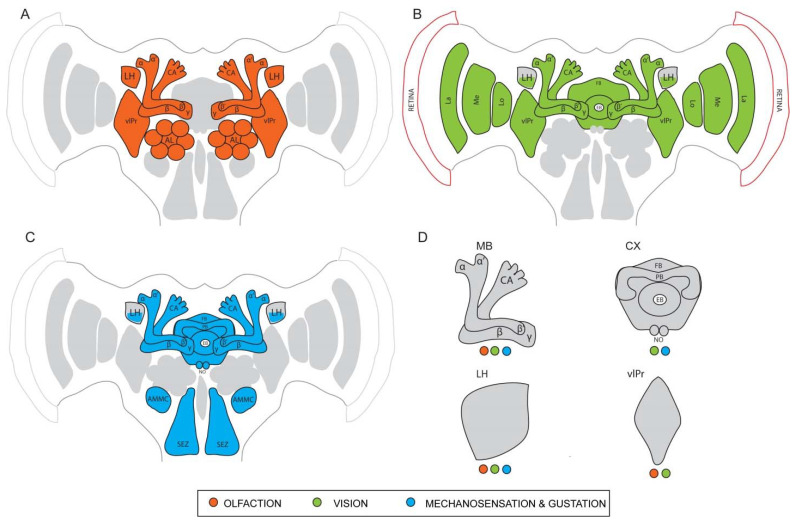
Schematic illustration of the adult *D. melanogaster* brain highlighting different brain centers. (**A**) Primary and higher olfactory centers that receive and process odor information. AL—antennal lobe; α, α’, β, β’, and γ—different lobes of the mushroom body (MB); CA—calyx of the MB; LH—lateral horn; vlPR—ventrolateral protocerebrum. (**B**) Primary visual centers and higher brain regions that receive and process visual input (color, light intensity and pattern). La—lamina; Me—medulla; Lo—lobula; FB—fan-shaped body of the central complex (CX); EB—ellipsoid body of the CX; NO—Noduli of the CX; (**C**) Primary and higher centers that receive and process gustatory and mechanosensory information. SEZ—sub-esophageal ganglion; AMMC—antennal mechanosensory and motor center; NO—noduli of the CX. (**D**) Higher brain centers like the MB, LH, CX, and the vlPR that are involved in multisensory processing with their respective inputs.

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
