# Peer review of "Multimodal Information Processing and Associative Learning in the Insect Brain"

_insects, 2022, doi:10.3390/insects13040332_

Round 1

Reviewer 1 Report

Thiagarajan & Sachse summarize behaviors across insect species that rely on multisensory integration and review the recent findings in terms of underlying neural circuitry, mostly based on studies in the fruit fly, Drosophila melanogaster. As natural environments are noisy and diverse, and animals are constantly exposed to multisensory stimulation, the approach to investigate multisensory integration is indeed necessary to understand complex brain functions.

The review is very relevant for the field and hopefully stimulates other researchers to test behaviors in more complex multisensory environments and also to investigate so far neglected neuropils and brain connections.

General:

I would suggest a reordering of the chapters. The first chapters could only be about behavior, including first the multisensory innate behaviors across species (current 1. And beginning of 4., also many innate behaviors mentioned in 5.) and then a description of multisensory input during learned behaviors (currently 5.), so that the first part focuses on the behavior. Then I would introduce the sensory circuits and together with it describe the studies that investigate neural circuits underlying multisensory integration. As the manuscript stands now, the sensory system neural circuits are introduced in 2./3., but then only mentioned again in 6. I think it would help the reader to have these parts together.

The authors mention that the MB is the first site of integration (line 531), which is probably not the case. Please mention/discuss that multisensory integration can not only happen in prominent higher brain centers, but also in early sensory processing systems. There is evidence in fly larvae based on connectomics (Berck et al., 2016 – AL neurons receive non-olfactory sensory input) and for example a very recent biorxiv publication (2022 Kazama – GRs respond to odorants, Odors drive feeding through gustatory receptor neurons in Drosophila | bioRxiv).

In general, the authors could highlight that the LH has traditionally not been seen as a flexible higher brain center. This review shows clearly that the LH is an additional plastic higher brain center next to the MB and CC. This might be also added to the conclusion.

Minor comments:

  • When generally describing the visual system, the authors do not go beyond the lobula, and lobula plate, I would suggest also describing the existence of the AOTU, the optic glomeruli, and eventually that a lot of visual input is processed in the Central Complex. This would complement the description of the olfactory system, extending to the higher brain regions MB and LH, and also fit the colorized scheme in Figure 1.
  • I would also suggest putting the description of the other sensory modality pathways into an extra paragraph (Line 140 and following).
  • The first part of the 4th chapter describes again general innate multisensory behaviors, such as feeding across insect species. This part could be included in the introduction. I would suggest focusing here on the contextual learning behavior part.
  • For gustatory input to the MB, I would suggest also citing this paper from the Scott lab: Kirkhart & Scott, 2015). They show that KCs respond to gustatory cues.
  • Line 551: For the representation of visual and olfactory cues in KCs: Vogt et al., 2016 showed that different sets of gamma KCs code for visual and olfactory cues, respectively. Vogt et al., 2014 and Aso et al., 2014 show that appetitive and aversive visual and olfactory learning required the same set of dopamine neurons and also overlapping and different sets of MBONs (see line 542).
  • For the functional connections between MB and CC, the authors might also cite Brembs, 2009. This paper shows that MBs are required for a specific type for visual learning and suggests interactions between the two neuropils (see line 630).
  • Line 604: for the bump in the EB, I would suggest citing Selig and Jayaraman et al., 2015, Nature.
  • Specific connections between MBONs and LHNs have also been described and even functionally shown in the fly larva (Eschbach et al., 2021, eLife). (see line 688).
  • Some references could be mentioned for VR arenas from Andrew Straw's lab for visual cues so far, or Matthieu Louis lab for optogenetics olfactory assays. Recently also a 3D VR setup for honeybee visual learning has been published by Martin Giurfa´s lab.
  • Unclear what is the reference for this sentence in line 677: This observation is further supported by the tethered flight experiment done in the same study, where the optogenetic activation of the LHNs drove the flies to move towards a visual stimulus.

Reviewer 2 Report

Overall this is a very nice review of the structure of the insect brain and regions implicated in processing of both single and multiple modalities.  The scope of the review is quite broad, so of course not every detail can be covered.  Nonetheless it is very thorough and does a good job especially of highlighting findings from multiple insect models.

minor comments: 

~line 64: in discussing gustation I am surprised the authors do not mention leg gustatory receptors

line 80: are there not some exceptions to the one glomerulus = one receptor rule?

line 94: "One of their major targets is the lateral horn" this implies I think that the MB is upstream of the LH while I think the literature more supports the idea that they are parallel structures, especially as AL DNs carry information both to the MB and to the LH.

~line 153: I think there are more studies of central auditory and wind processing for mechanosensation that could be mentioned here beyond the AMMC.  (e.g. WED, LAL, VLP, ATL)

line 355: could perhaps mention the integration of wind (from mechanosensation and optic flow) with odor as another prominent multisensory behavior observed across insects.

conclusions:  It might be helpful if the authors outlined key questions they think the field should address in the next years concerning multi sensory integration and representation.

Reviewer 3 Report

General comments:

I enjoyed reading this is interesting and timely review, taking recent advances in multimodal integration of sensory signals and their involvement in learning and memory into account. The manuscript is well-structured, but there are some points which could be improved in my eyes. The manuscript is written in a sophisticated language, but word-use often does not completely match the meaning as far as I understand. I indicate several cases in the detailed evaluation below, but without being exhaustive. I think it would be worth-while to carefully work through the language of the text again, before re-submitting.

In parts 3. and 4. you describe the insect olfactory and visual system (that is what you indicate in the title), but then really concentrate very quickly on D. melanogaster. Because you describe later nevertheless quite a few data on other insects for multimodal integration in different brain parts, maybe you could keep the descriptions of each system slightly more general? I am not against keeping D. melanogaster as main model and use It for illustrations, but a bit more important general part might be useful.

For some parts, maybe not all introductory generalities are necessary to understand what you develop behind (e.g. 6.1, 6.2, 6.3, and details on split-Gal4 technique lines 658ff, the technique has been mentioned before already).

I am not very convinced about the usefulness of Figure 2: all information contained in the brain scheme is already provided in Figure 1, is it necessary to show it again (and that big?)? What do you want to implicate with the green/yellow color of the structures you highlight in the general brain scheme? It is different from the modality color code, still two-colored… The color code for the modalities is good to maintain from Fig. 1, but the dots underneath the general structures without being more precise do not provide a whole lot of information…

Throughout the text:

Drosophila melanogaster at first occurrence, later use D. melanogaster

Detailed comments:

Line 31/32: reformulate, suggestion: Such multimodal integration has also been investigated in the context of….

Line 37: Please reformulate: there are no multimodal behaviors (behaviors can be elicited by multimodal cues)

Lines 40/41: awkward sentence, please reformulate

Line 45: the word “modalities” is not adapted here (you don’t describe the anatomy of the modality but the anatomy of the sensory systems)

Line 49: same as above, behaviors can be influenced by several modalities, but behaviors are not multimodal

Line 51: …in the central nervous system of insects with emphasis…

Line 78: …act as the first synaptic site for OSNs.

Line 96: replace “naïve” by “innate”

Line 102: Visual cues are perceived…

Lines 119-121: awkward sentence, please reformulate (largest proportion instead of repertoire?)

Line 122: …flow with…

Line 137: … preliminary visual information…

Before Line 140, please add an additional sub-title (this is no longer the visual system)

Line 151: Antennal mechanosensory and motor center (not cortex)

Line 175: please reformulate: there is no cross-modal behavior (just take out cross-modal?)

Line 192: consider improving this sentence: The more numerous the cues…, the quicker…

Line 201: …unconditioned stimulus…

Line 308: …and makes the ant remember…

Line 323: … a strong recognition of…

Line 350: again, behaviors are not sensory, please reformulate

Lines 360 -362: it is confusing to use the terms parasitoid and predator for the same natural enemy (if it is a parasitoid, it is not normally considered as a predator)

Line 440/41: reformulate, suggestion: Flies still showed this behavior after targeted blocking of…

Line 455: …depends…

Line 471: …two major ganglion groups…

Line 473: … show…

Line 488 …major multi-sensory integration centers…

Figure 2 legend: explain abbreviations of the different parts of the central complex

Line 525: …observed in hymenopteran models..

Line 527: either: to mostly integrate information received through two kinds of visual PNs, or: was shown to mainly receive input from two kinds…

Lines 528-532: please revise logic of the sentence (parallel functioning versus convergence): information from parallel pathways can be grabbed by higher order neurons spanning the different areas (as illustrated by your next sentence on crickets: isn’t this also a MBON, as what you describe in the next paragraph?)

Line 560: …upon which motor (or locomotion?) behaviors…

Lines 572-574: The perspectives of this part would gain to be less vague: (e.g. something like by knocking out specific neuron types in the MBs, their role in multimodal integration could be tested in behavioral paradigms…)

Line 576: take out comma

Line 642/643: again, it is not the behavior which is multi-modal, but you mean behavior influenced by different sensory modalities

Line 645: all structures you described are part of the protocerebrum, so rather say: …another part of the protocerebrum which has garnered major interest…

Line 646: …the two major higher…

Line 649: …antennal lobes…

Round 2

Reviewer 3 Report

I thank the authors for the nice revision of the manuscript, which reads very well now. There are a few typos left and species names are not always in italics (in the text and especially in the reference list).